# Comparison between Capillary and Serum Lactate Levels in Predicting Short-Term Mortality of Septic Patients at the Emergency Department

**DOI:** 10.3390/ijms24119121

**Published:** 2023-05-23

**Authors:** Matteo Guarino, Benedetta Perna, Alice Eleonora Cesaro, Michele Domenico Spampinato, Rita Previati, Anna Costanzini, Martina Maritati, Carlo Contini, Roberto De Giorgio

**Affiliations:** 1Department of Translational Medicine, St. Anna University Hospital of Ferrara, University of Ferrara, 44124 Ferrara, Italy; grnmtt@unife.it (M.G.);; 2Emergency Department, St. Anna University Hospital of Ferrara, 44124 Ferrara, Italy; 3Department of Clinical Sciences, Infectious and Dermatology Diseases, St. Anna University Hospital of Ferrara, University of Ferrara, 44124 Ferrara, Italy

**Keywords:** capillary lactates, mortality, sepsis, septic shock, serum lactates

## Abstract

Sepsis is a time-dependent and life-threating condition related to macro- and micro-circulatory impairment leading to anaerobic metabolism and lactate increase. We assessed the prognostic accuracy of capillary lactates (CLs) vs. serum ones (SLs) on 48-h and 7-day mortality in patients with suspected sepsis. This observational, prospective, single-centre study was conducted between October 2021 and May 2022. Inclusion criteria were: (i) suspect of infection; (ii) qSOFA ≥ 2; (iii) age ≥ 18 years; (iv) signed informed consent. CLs were assessed with LactateProTM2^®^. 203 patients were included: 19 (9.3%) died within 48 h from admission to the Emergency Department, while 28 (13.8%) within 7 days. Patients deceased within 48 h (vs. survived) had higher CLs (19.3 vs. 5 mmol/L, *p* < 0.001) and SLs (6.5 vs. 1.1 mmol/L, *p* = 0.001). The best CLs predictive cut-off for 48-h mortality was 16.8 mmol/L (72.22% sensitivity, 94.02% specificity). Patients within 7 days had higher CLs (11.5 vs. 5 mmol/L, *p* = 0.020) than SLs (2.75 vs. 1.1 mmol/L, *p* < 0.001). The multivariate analysis confirmed CLs and SLs as independent predictors of 48-h and 7-day mortality. CLs can be a reliable tool for their inexpensiveness, rapidity and reliability in identifying septic patients at high risk of short-term mortality.

## 1. Introduction

Sepsis is a time-dependent and life-threatening condition burdened by a high mortality rate if not early recognized and treated [1,2]. It affects over 2 million cases/year worldwide with a mortality rate of 20–40% estimated to grow steadily in the next years [3,4]. According to Singer et al., sepsis is defined as a life-threatening organ dysfunction caused by a dysregulated host response to infection and should be suspected in patients with an infective source. In these subjects a quick Sequential Organ Failure Assessment (qSOFA) should be assessed and, if ≥2, indicates patients at higher risk of in-hospital death. The diagnosis is confirmed in case of a Sequential Organ Failure Assessment (SOFA) score ≥ 2. Septic shock is defined by the need of a vasopressor to maintain a mean arterial pressure (MAP) ≥ 65 mm Hg and serum lactate level ≥ 2 mmol/L [1]. However, the 2021 guidelines recommended against the use of qSOFA as a single screening tool, suggesting the use of National Early Warning Score (NEWS) or systemic inflammatory response syndrome (SIRS) for their better sensitivity vs. qSOFA in predicting patient’s outcome [2]. Although only partially understood, the current knowledge indicates that the pathophysiological mechanisms involved in sepsis include the concurrence of both pro-inflammatory and anti-inflammatory mediators [5,6,7]. In this context, the cytokine cascade (mainly triggered by interleukin 1-β and 18), responsible for sepsis clinical manifestations, occurs as a result of a number of events leading to a dysregulated host response to infections and the subsequent multi-organ failure [1,7]. Furthermore, a reduced perfusion of vital organs is the key mechanism leading to shock, which compromises both macro- and micro-circulation and leads to to lactic acid production [8,9,10,11]. However, lactate elevation may be related to a wide range of mechanisms: (i) poor tissue perfusion or hypoxia; (ii) underlying disease processes (e.g., malignancy, liver/renal failure, diabetic and alcoholic ketoacidosis); (iii) drugs or toxins; and (iv) congenital errors of metabolism (e.g., pyruvate dehydrogenase deficiency, glucose-6-phosphatase deficiency, congenital mitochondriopathies) [8,9].

Several studies demonstrated that the elevation of lactates is an independent predictor of mortality in sepsis [11,12,13,14,15,16]. Specifically, lactate levels raising from 2 to 4 mmol/l leads to a 28-day mortality rate increase close to 15% in patients without hypotension [1,16]. Moreover, the mortality rate is higher as lactate levels increase [2]. Liu et al. demonstrated that high levels of serum lactate (SL) should be considered an early alarm indicator of multiorgan failure and a more accurate tool than qSOFA/SOFA in predicting mortality in patients with sepsis/septic shock [11]. Conversely, the reduction in SL indicates the restoration of tissue perfusion and predicts a decreased fatal outcome [16,17]. Arterial and venous blood samples have the same accuracy for lactate levels < 4 mmol/L, whereas, venous assays above this cut-off seem to have a better sensitivity in the early identification of sepsis [17,18,19].

While macro-circulation damage has been widely studied, micro-circulatory abnormalities (i.e., decreased capillary density, decreased proportion of perfused capillaries and increased heterogeneity of blood flow) have been only recently analysed and partially understood. A number of factors (including hypoxia, hypoperfusion and bacterial toxins) can impair the capillary endothelium causing what is referred to as “glycocalyx shedding” [20,21,22,23]. This condition is known to damage micro-circulation, thus worsening tissue perfusion and leading to anaerobic cell metabolism with lactic acid production and shock [6,8,9,10,11]. Since the persistence of microcirculatory changes is associated with poor outcomes [9], recent studies proposed “micro-circles resuscitation” as one of the main targets in sepsis treatment [24,25,26]. Various clinical and instrumental tools have been proposed to assess microcirculatory dysfunction [27,28,29,30]; however, despite multiple attempts, there is no consensus on the best method. Capillary lactate (CL) assay, obtainable through inexpensive and handheld devices, has been proposed to achieve this goal, although with variable results and an unclear role [27,31,32]. CL (and the related cut-off) has not been validated yet, despite different studies showed a high correlation with SL [27,33,34,35,36].

The objective of this study was to determine the prognostic accuracy of CL measurement with a point-of-care (POC) analyzer vs. SLs on 48-h (primary endpoint) and 7-day mortality (secondary endpoint) in patients with suspected sepsis admitted to the Emergency Department.

## 2. Results

A total of 203 patients (43.3% males) with a median age of 85 (IQR 74–90) years met the inclusion criteria and 188 (92.4%) had a confirmed final diagnosis of sepsis. Nineteen (9.3%) patients died within 48 h from ED admission, while 28 (13.8%) within 7 days. Sepsis etiologies have been categorized into five groups, i.e., respiratory (n = 53, 28.2%), urinary (n = 79, 42.0%), abdominal (n = 12, 6.4%), miscellaneous (n = 21, 11.2%) and indeterminate (n = 23, 12.2%). NEWS (10 vs. 8, *p* = 0.001), SOFA score (8 vs. 4, *p* < 0.001), CLs (19.3 vs. 5, *p* < 0.001) and SLs (6.5 vs. 1.1, *p* = 0.001) were higher in patients who died within 48 h from admission. No differences in terms of age, sex, CCI, fluid treatment or antibiotics administration were observed between the two groups. Further results are described in Table 1. There was a significative medium correlation between CLs and SLs (0.497, *p* < 0.001), NEWS score (0.359, *p* < 0.001) and SOFA score (0.412, *p* < 0.001) (Appendix A). No correlation was observed between CLs and SBP or MAP, while SLs showed a slightly negative correlation with SBP (−0.253, *p* = 0.001). The best CLs cut-off predictive of death within 48 h was 16.8 mmol/L, with 72.22% sensitivity (95% CI 46.5–90.3), 94.02% specificity (95% CI 89.6–97), 12.08 +LR (95% CI 9.1–16.1), 0.3 -LR (95% CI 0.1–0.8), 54.2% PPV (95% CI 32.8–74.4) and 97.2% NPV (95% CI 93.6–99.1). A multivariable analysis including NEWS score, SOFA score, CLs ≥ 16.8 mmol/L and SLs ≥ 2 mmol/L demonstrated that CLs and SLs are strong independent predictors of poor outcome (death) within 48 h, with OR equal to 22.9 (95% CI 4.7–111.5, *p* < 0.001) and 13.8 (95% CI 1.4–132.9, *p* = 0.023), respectively (Table 2). Patients who died in the first 7 days had lower GCS (9 vs. 13, *p* = 0.001), higher NEWS (10.5 vs. 8, *p* < 0.001), SOFA score (7 vs. 4, *p* < 0.001), CLs (11.5 vs. 5 mmol/L, *p* = 0.020) and SL (2.75 vs. 1.1 mmol/L, *p* < 0.001) (Table 1). Considering the cut-off obtained for the primary endpoint, a CLs ≥ 16.8 mmol/L predicted a poor outcome within 7 days with 42.9% sensitivity (95% CI 24.5–62.8), 96.64% specificity (95% CI 88.9–96.8), 6.74 +LR (95% CI 3.3–13.8), 0.6 -LR (95% CI 0.4–0.8), 52.2% PPV (95% CI 34.8–69) and 91% NPV (95% CI 88–93.3). Figure 1 and Figure 2 illustrate Fagan’s nomograms of CLs ≥ 16.8 mmol/L and SLs ≥ 2 mmol/L on 48-h and 7-day mortality, respectively. A multivariable model including NEWS score, SOFA score, CLs ≥ 16.8 mmol/L and SL ≥ 2 mmol/L showed that both CLs and SLs were strong independent predictors of poor outcome within 7 days (Table 2). Spearman’s correlation coefficients (SCCs) are summarized in Appendix A highlighting a positive correlation between CLs/SLs (SCC 0.497, *p* < 0.001), a negative one between SLs/systolic blood pressure (SBP) (SCC −0.253; *p* = 0.001) and no significant correlation between CLs/SBP (SCC −0.105 *p* = 0.166).

Sensitivity analysis on the primary outcome has been reported in the Appendix A. The main findings were consistent among patients <80 and ≥80 years.

The Bland-Altman method shows that 96% of the differences between CLs and SLs assays are within the 95% limits (191 out of 199) (Appendix A). Furthermore, the comparison among CL, SL, NEWS and SOFA in assessing 48-h and 7-day mortality have been plotted in Appendix A (panel A and B). No differences were shown at the pairwise comparison of the areas under the curves (AUCs) of CLs vs. SLs (*p* = 0.464).

## 3. Discussion

The data of the present study showed that CLs are highly predictive of both 48-h and 7-day mortality, regardless SL and blood pressure. Furthermore, compared to SLs ≥ 2 mmol/L, CLs ≥ 16.8 mmol/L better identifies patients at high risk of 48-h (+LR of 12.08 vs. 3.68 and OR 22.9 vs. 13.8) and 7-day mortality (+LR of 6.74 vs. 3.68 and OR 6.09 vs. 3.88) for sepsis/septic shock. Numerous studies, evaluating the efficacy of CLs in predicting mortality, yielded heterogeneous results [37,38,39,40]. In a prospective, multicenter, observational analysis on 941 patients, López-Izquierdo et al. demonstrated a poor correlation between CLs and 30-day all-cause mortality [37]. While acknowledging the recruited large sample size, we would like to highlight that the study was performed on adult patients with heterogeneous acute diseases and the outcome considered was 30-day mortality. In contrast, our analysis considered short-term mortality testing whether CLs, as an index of the underlying micro-circulation damage, might better predict the poor outcome.

In 2017, a systematic review evaluated the prognostic ability of SL/CL POC assays in suspected septic patients at first medical contact. This study concluded that there is no high-quality evidence to support the use of POC lactate in community settings [38]. However, three out of the six studies included in this review reported a significant correlation between POC-SLs and in-hospital mortality [41,42,43]. Only one study, considering the ability of CLs in predicting fatal outcome [39], concluded that although not significant a POC lactate assay might help in assessing the mortality risk. This analysis had several limitations, including a small sample size (n = 59) and unclear patient inclusion criteria. In 2015, Contenti et al. proposed a comparison between arterial, venous, and capillary blood lactates in a cohort of severe septic patients suggesting that all these assays might predict 28-day mortality [40]. The authors indicated that venous, rather than arterial and capillary, samples could be more effective to early detect severe sepsis. However, this manuscript was limited by a low number of patients (n = 103) and the use of “severe sepsis” criteria that were completely abolished by the Surviving Sepsis Campaign (SSC) [1]. Manzon et al. concluded that CLs at the ED admission were significantly higher in patients with 28-day mortality vs. survivors (5.7 vs. 2.9 mmol/L, *p* = 0.003), being as useful as mottling score and capillary refill time (CRT) in identifying patients at higher risk of death [44].

Shock is defined as a state of tissue hypoperfusion instead of a hypotension condition [45]. In this context, the poor perfusion may lead to macro- and micro-circle damages, thus elevating lactate levels [9]. Markers of micro-circulation injury (e.g., mottling or CRT) are as important macro-circulatory dysfunction markers (e.g., MAP or cardiac output) in guiding resuscitation measures of patients with septic shock [24], being able to identify persisting hypoperfusion even though systemic hemodynamic variables have been restored [46]. In addition, peripheral perfusion restoration was more effective than SL clearance in reducing 28-day mortality in septic shock patients [47,48]. The analysis performed in this study showed a positive correlation between CLs and SLs, confirming that when CLs increase, also SLs increase. Furthermore, a negative correlation between SLs and SBP has been identified, whereas there was no correlation between CLs and SBP, supporting that SLs increase in hypotensive states while CLs may increase before SBP values decrease. The Bland-Altman plot showed that the difference between CL and SL increases (in parallel with lactate level), reaches the peak and then decreases (Appendix A). This pattern indicates that the occurrence of microcirculatory impairment causes a CL increase without a concomitant one in SL. When the patient’s circulatory functions deteriorate, SL increases leading to a reduction in the difference between CL and SL. This phenomenon suggests that CL is able to detect early states of microcirculatory dysfunction not yet associated with SL increase. Although CL and SL showed a significant moderate correlation, both these assays were found to be independent predictors of death within 48 h and 7 days. This main result is in accordance with other studies, confirming that signs of microcirculatory dysfunction can detect a subgroup of septic patients at higher risk of fatal outcome [49,50,51].

We would like to acknowledge some limitations of our study. First, it is a single-center analysis with a small sample size, which considerably reduced the statistical power of this investigation. Second, since many different conditions can exhibit peripheral hypoperfusion leading to lactate level increase, a thorough clinical definition of the actual underlying disease is deemed necessary before a diagnosis of sepsis is applied [9,52]. Third, only one measurement of CLs and SLs had been performed for each patient (with no possibility to assess lactate clearance for further prognostic information). Fourth, we did not correlate CLs and SLs with CRT/mottling score which may provide further information about patients’ perfusion [53,54].

On the other hand, the main strengths of this study include: First, this is a prospective analysis on the role of CLs conducted at the ED in patients with suspected sepsis. Second, we addressed the low invasiveness, rapidity, reliability and relative inexpense method of capillary sampling compared to the ABG. Since sepsis is a time-dependent and life-threatening condition, a tool that can early identify patients with a potential risk of fatal outcome is very useful for emergency physicians [55]. Third, the lack of an early CLs assay detection by physician who treats the enrolled patient prevents possible biases in the management. Indeed, the knowledge of a high CL value might induce the physician to overtreat the patient (e.g., early vasopressor administration or more aggressive fluid replacement [56]), thus probably affecting the outcome.

## 4. Materials and Methods

This is an observational, prospective, monocentric and non-interventional study conducted between October 2021 and May 2022 at the Emergency Department of St. Anna University Hospital in Ferrara, a large third-level centre with >70,000 patients visited per year. Inclusion criteria were: (i) clinical suspect of infectious disease; (ii) qSOFA ≥ 2; (iii) age ≥ 18 years; (iv) a signed informed consent was obtained from each involved patient (or their relatives in case of overall severe clinical conditions). This analysis was conducted as a single-blinded study since the clinician who treated the enrolled subject did not know the result of the capillary sample. Since the present paper has been based on the 2016 guidelines, the screening tool used for the early suspect of sepsis/septic shock was the qSOFA.

The following data were considered in each subject: (i) capillary and arterial blood samples to estimate lactate levels; (ii) vital parameters (i.e., systolic and diastolic blood pressure, heart rate, respiratory rate, oxygen saturation, body temperature and Glasgow Coma Scale); (iii) laboratory tests (i.e., blood cells count, C-Reactive Protein, creatinine, electrolytes, bilirubin, alanine aminotransferase and coagulation parameters); (iv) pharmacological treatment during hospitalization; (v) occurrence of 48-h and 7-day mortality; (vi) final diagnosis. Comorbidities were assessed using Charlson Comorbidity Index (CCI). CLs were assessed on admission at ED with a specific instrument (i.e., LactateProTM2^®^, Arkray Global Business Inc., Kyoto, Japan) commonly used in sport medicine [23] and proposed to measure lactate clearance in intensive care unit (ICU) and at the ED [24,25]. The LactateProTM2 is a handheld POC analyzer which assesses CLs levels by enzymatic amperometric detection. Indeed, blood lactates react with the reagent on the test strip, thus producing a small electrical current proportional to the concentration of lactates. For a single determination, the device assesses a blood sample of 0.3 µL through specific strips and obtains the result in 15 s. The measurement range is between 0.5–25.0 mmol/L [23]. All samples were compared to those obtained simultaneously by arterial blood gas (ABG) samples determined through the ABL90 Flex^®^ blood gas analyzer (Radiometer Medical ApS, Brønshøj, Denmark) which provides the result in 35 s. The study was approved by the Ethics Committee of the Area Vasta Emilia Centro (CE-AVEC) (protocol n. 447/2021/Disp/AOUFe).

### Statistical Analysis

Categorical data were expressed as absolute frequencies and percentages, while medians and interquartile range (IQR) were reported for continuous variables. Differences between patients deceased or discharged for sepsis were compared with Pearson’s X2, student *t*-test and Mann-Whitney tests as appropriate. The correlation between two independent parametric variables measured on a continuous scale was determined using Pearson’s correlation in case of normal distribution and Spearman rank correlation in case of not normal distribution. According to Cohen, the r absolute value was considered little, medium, or high if equal or larger than 0.1, 0.3 and 0.5, respectively. The association between 48-h and 7-day mortality and the investigated tools (i.e., CLs and SLs) was studied using univariate and multivariable logistic regression analysis. Odds ratios (ORs) and their 95% confidence intervals (CI) were reported. The best cut-off was identified according to the highest Youden index. Sensitivity, specificity, negative predictive value (NPV), positive predictive value (PPV), negative likelihood ratio (−LR) and positive likelihood ratio (+LR) were calculated for each cut-off. The Bland-Altman method was used to determine the concordance of CL and SL measurements [57]. The discrimination ability of each score was evaluated via the area under the receiver operating characteristic (AUROC). The AUROCs of the scores were compared via the DeLong test [58]. The Statistical Product and Service Solution (SPSS) 23.0 for Windows (IBM Corp., Armonk, NY, USA) and MedCalc^®^ Statistical Software version 19.8 (MedCalc Software Ltd., Ostend, Belgium) were used for statistical analyses; the significance level was set for *p* < 0.05.

## 5. Conclusions

In this single-centre and prospective study, we highlighted that CLs can be a reliable tool for the early identification of a subset of septic patients at higher risk of 48-h and 7-day mortality. Further confirmatory results on larger series are eagerly awaited to aid emergency physicians in establishing timely outcome of septic patients.

## Figures and Tables

**Figure 1 ijms-24-09121-f001:**
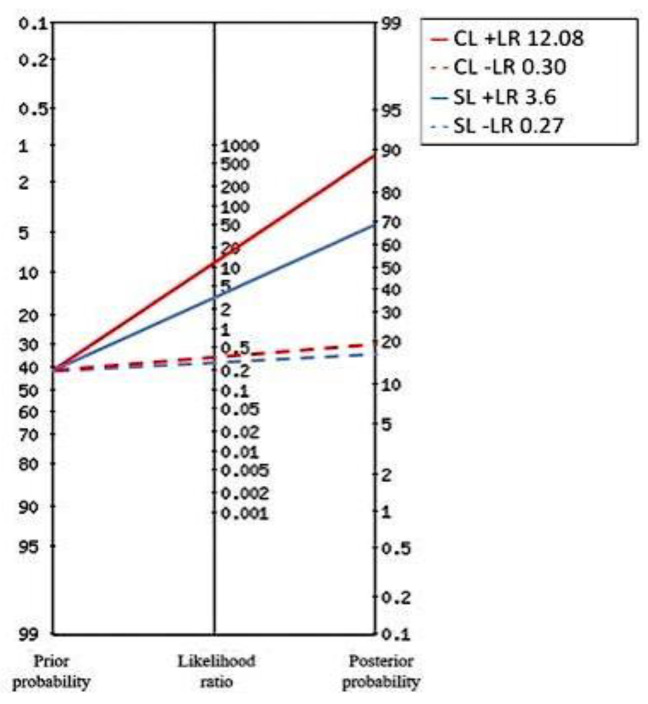
Fagan’s nomograms illustrating the effect of CLs ≥ 16.8 mmol/L and SLs ≥ 2 mmol/L on 48-h mortality.

**Figure 2 ijms-24-09121-f002:**
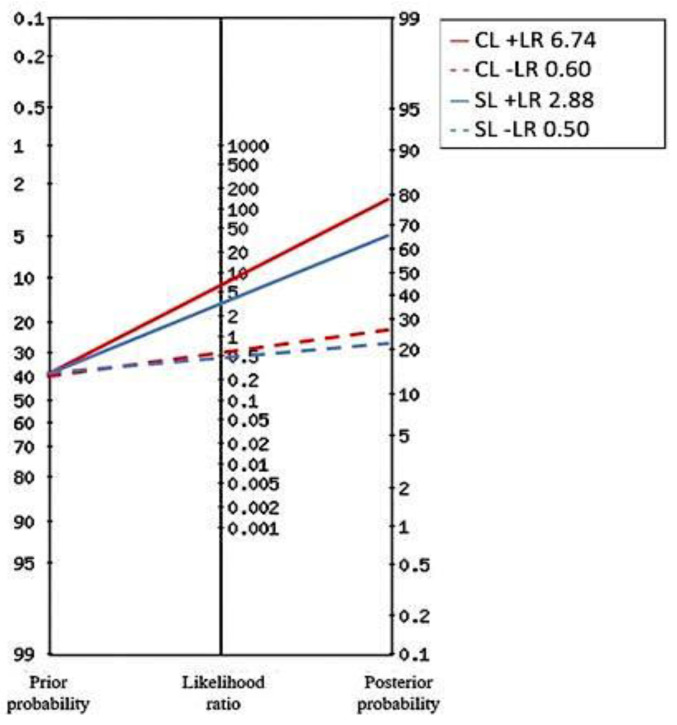
Fagan’s nomograms illustrating the effect of CLs ≥ 16.8 mmol/L and SLs ≥ 2 mmol/L on 7-day mortality.

**Table 1 ijms-24-09121-t001:** Clinical and laboratory features of enrolled patients.

Clinical/Lab Features	48-h Mortality			7-Day Mortality		
	Survived	Deceased		Survived	Deceased	
	n = 184 (90.7%)	n = 19 (9.3%)	*p*	n = 175 (86.2%)	n = 28 (13.8%)	*p*
Male	79 (42.9)	10 (52.6)	0.395	76 (43.4)	11 (39.3)	0.68
Age, years	85 (79–89)	85 (73–90)	0.74	85 (79–90)	84 (71–88)	0.91
CCI > 4	36 (19.4)	5 (26.3)	0.47	34 (19.4)	6 (21.4)	0.8
Admission to the ED						
MAP	76 (60–92)	78 (55–92)	0.55	77 (62–92)	75 (56–90)	0.51
Altered mental status	152 (82.6)	18 (94.7)	0.54	142 (82.1)	27 (96.4)	0.06
GCS	13 (11–14)	9 (7.5–13)	0.001	13 (10.5–14)	8 (6–11)	0.01
Laboratory data						
Hb g/dL	11.5 (9.65–13.3)	11 (9.5–13)	<0.001	11.5 (9.65–13.3)	10.9 (8.3–13)	0.23
PLT (×1000/mmc)	224 (165–281)	230.5 (150–320)	0.84	230 (170–286)	187 (144–287)	0.47
Bilirubin mg/dL	0.8 (0.52–1.19)	0.79 (0.56–1.06)	0.93	0.795 (0.52–1.18)	0.87 (0.51–1.06)	0.94
INR	1.15 (1.07–1.26)	1.32 (1.15–1.69)	0.01	1.16 (1.07–1.27)	1.34 (1.2–1.8)	0.03
C-reactive protein mg/dL	10.1 (3.17–16.17)	13.47 (2.82–22.07)	0.46	10. (3.17–15.98)1	14.39 (2.28–23.47)	0.78
Urea mg/dL	65 (40–110)	95 (53–173)	0.06	64 (40–110)	121 (73.5–244.5)	0.02
Creatinine mg/dL	1.24 (0.83–1.92)	1.92 (0.96–3.63)	0.12	1.24 (0.83–1.92)	2.86 (1.32–4.57)	0.02
DTX mg/dL	128 (109–166.5)	163 (123–326)	0.06	128 (110–170)	170.5 (123.5–276)	0.17
Blood Gas Analysis						
pH	7.45 (7.4–7.48)	7.37 (7.18–7.40)	<0.001	7.45 (7.4–7.48)	7.26 (7.16–7.37)	<0.001
pO_2_ mmHg	74.3 (66–87.75)	86 (68–168)	0.18	74.5 (66–88)	91.35 (66.5–127)	0.44
pCO_2_ mmHg	35.7 (32–41)	35.5 (30.6–40)	0.93	35.8 (32.4–41)	32 (24.6–38)	0.5
HCO_3_ mmol/L	25.8 (22.5–28.5)	19 (12.7–26.3)	0.12	25.8 (22.4–28.6)	15.25 (11–19)	0.01
P/F	334.5 (269–389)	374 (210–446)	0.61	339.5 (274–393)	231 (113–446)	0.65
Severity scores						
SIRS ≥ 2	124 (67.8)	16 (84.2)	0.14	116 (67.4)	23 (82.1)	0.12
NEWS ≥ 7	111 (59.7)	17 (89.5)	0.11	103 (58.9)	23 (82.1)	0.02
NEWS	8 (5–10)	10.5 (9–12)	<0.001	8 (5–10)	10 (9–13)	0.001
SOFA ≥ 2	154 (82.8)	17 (89.5)	0.45	146 (83.4)	24 (85.7)	0.76
SOFA	4 (3–6)	7 (5–8)	<0.001	4 (3–6)	8 (7–9)	0.001
Treatments in the ED						
O2 supplementation	78 (42.2)	13 (72.2)	0.01	73 (42)	17 (63)	0.04
Optimal fluid replacement	127 (68.6)	16 (84.2)	0.16	119 (68.4)	22 (78.6)	0.27
Antibiotics administration	95 (51.9)	12 (66.7)	0.23	89 (51.7)	17 (63)	0.28
Vasopressors administration	4 (2.2)	3 (15.8)	0.01	4 (2.3)	3 (10.7)	0.02
Serum and Capillary Lactate						
SLs mmol/L	1.1 (0.7–2)	2.75 (1.55–8.1)	<0.001	1.1 (0.7–2)	6.5 (2.4–11.1)	0.001
SLs ≥ 2 mmol/L	47 (25.3)	15 (78.9)	<0.001	43 (24.6)	18 (64.3)	<0.001
CLs mmol/L	5 (3.1–9.1)	11.15 (3.7–22.95)	<0.001	5 (3–9.05)	19.3 (10.8–25)	<0.001
CLs ≥ 16.8 mmol/L	11 (5.9)	13 (68.4)	<0.001	11 (6.3)	12 (42.9)	<0.001

Footnote: The table expresses median values (IQR) or numerosity (percentage) as appropriate.

**Table 2 ijms-24-09121-t002:** Multivariate analysis including CLs, SLs, SOFA and NEWS score.

	48-h Mortality			7-Day Mortality		
	OR	95% CI	*p*	OR	95% CI	*p*
CLs ≥ 16.8mmol/L	22.960	4.72–111.53	<0.001	6.090	1.78–20.87	0.004
SLs ≥ 2 mmol/L	13.790	1.43–132.9	0.020	3.880	1.25–12.05	0.020
SOFA	1.460	0.98–2.18	0.060	1.160	0.91–1.48	0.210
NEWS	1.030	0.77–1.38	0.830	1.130	0.94–1.36	0.160

Footnote: Hosmer and Lemeshow test X-squared = 2.51, Degree of Freedom = 8, *p* = 0.961.

## Data Availability

The datasets generated and/or analysed during the current study are not publicly available due to privacy policy but are available from the corresponding author on reasonable request.

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
