# Peer review of "Comparison between Capillary and Serum Lactate Levels in Predicting Short-Term Mortality of Septic Patients at the Emergency Department"

_ijms, 2023, doi:10.3390/ijms24119121_

Round 1

Reviewer 1 Report (New Reviewer)

I read with great interest the manuscript by Guarino et al. on the use of capillary and serum lactate levels in predicting short-term mortality of septic patients at the Emergency Department. The paper is sound and interesting. However, I have some issues to be addressed.

- I do not really understand why some parts of the manuscript are colored in red.

- The introduction should be summed up as some concepts are redundant. For example, there are two definition of sepsis, the statement that "the mortality rate is higher as lactate levels increase" is repeated at the end of the introduction.

- In the "results" section, please remove all the information reported in the tables, in order to shorten the section and make it easier to read.

- Line 241-243. It should also be underlined that CRT is easy to perform (doi: 10.1186/s12871-022-01920-1) and widely used (doi: 10.1186/s12873-022-00681-x). However, you did not assess it during the study. Please discuss it in the limitation section and add these 2 references.

Since this is a single center observational study, please change the first sentence in the conclusion section in order to avoid the word "demonstrated".

Author Response

Reviewer 2 Report (New Reviewer)

Guarino et al, in their manuscript “Comparison between Capillary and Serum Lactate Levels in Predicting Short-term Mortality of Septic Patients at the Emergency Department,” evaluate the usefulness of capillary and serum (arterial) lactate levels in septic patients on ED presentation in predicting 48 hour and 7 day mortality in this patient population. In their group of 203 patients, both capillary lactate and serum lactate levels independently predicted 48 hour and 7 day mortality. This is an important study as POC testing of capillary lactate may provide an inexpensive way to rapidly identify patients with high risk for mortality due to sepsis.

My primary comments focus on the methods of testing. First, the authors state POC measurement of capillary lactate is inexpensive, but they do not actually describe the cost difference between capillary lactate measurement using the LactatePro TM2 and serum lactate measurement using the blood gas analyzer. Stating the actual relative costs between these two devices would strengthen their argument for using capillary lactate measurement based on expense.  Second, using two different devices to measure lactate raises the question of inter-device reliability. For example, if lactate were measured on the same serum blood sample in both the LactatePro and the blood gas analyzer, would the results be similar between the two devices? The presumption is that higher lactate levels in capillary blood versus arterial blood reflect a difference between the microcirculation and larger blood vessels, but could some of the difference be due to difference in measuring devices?

Figures 1 and 2 both state a prior probability of 40% for mortality; what is the basis for this number?

There are some language issues that can be addressed by copy editing.

Author Response

Reviewer 3 Report (New Reviewer)

In many Italian ED lactate can be measured by a point of care

Author Response

Reviewer 4 Report (New Reviewer)

The article Comparison between Capillary and Serum Lactate Levels in Predicting Short-term Mortality of Septic Patients at the Emergency Department assessed the prognostic value of capillary lactates (CLs) vs. serum ones (SLs) on 48-hour and 7-day mortality in patients with suspected sepsis. They have found that CLs can be a reliable tool for identifying septic patients at high risk of short-term mortality. 

Comments:

1.      I’m assuming that text in red is added in response to previous reviewer(s) comments. Changes are mainly made in the introductory part of the paper trying to set the stage for CLs being better predictor of septic short-term patient mortality vs. SL levels. The addition about micro-circulation damage and abnormalities in sepsis is appreciated! However, more should be added about increased lactate levels in sepsis, what are causes of the rise and its importance. Also, discussion should be expanded!

2.      As authors pointed out, many different conditions can exhibit peripheral hypoperfusion leading to increased CL levels, could authors suggest which combination of factors/measurements would further increase predictability of short-term mortality in septic patients?

3.      How many measurements of CLs and SLs and in which time intervals authors suggest should be done for each patient with suspected sepsis to increase prognostic value of the test?

Moderate editing of English language is needed.

Author Response

This manuscript is a resubmission of an earlier submission. The following is a list of the peer review reports and author responses from that submission.

Round 1

Reviewer 1 Report

Thank you for the opportunity to review this manuscript, which tries to elucidate the role capillary lactate levels measured in ED could play in predicting outcome. In a single centre study the authors evaluated 203 very elderly patients with clinical suspicion of infection and adverse signs for sepsis. They found that Capillary and serum lactate (CL and SL) shown a significant correlation, and that high CL was a predictor of mortality at 48 hours.

There are several major issues with the study, some methodological, some conceptual.

1. The authors wrongly state in the introduction, that the 2016 sepsis definitions said sepsis is suspected if there is a clinical suspiscion of infection and the qSOFA is >2. Sadly the qSOFA is the most misinterpreted tool and as the authors correctly state, for this reason it's use is discouraged. What Singer et al. said, was that patients with clinical suspicion of infection MAY be evaluated with a simple score, to put them into the high or low risk of death category at 28 days. Please refrain to try to assert that qSOFA is in any way part of the sepsis-3 diagnostic pathway.

2. I really struggle with the convoluted description on how CL levels could be "faster" and "earlier" markers of deterioration compared to SL. The authors hypothesise that this might be the case, offering no data to support the hypothesis. Then they proceed to evaluate if their hypothesis of CL being a better predictor in a hypothesised scenario is valid??? I do think they have put the cart before the horse here.

3. The prudent first step to investigate this would have been a time-series measurement and correlation of CL and SL using identical detection methods, such as an arterial blood gas analyser, which can process both CL and SL samples.

4. The investigators used a CL point-of-care method, which has been developed for non-healthcare use. At minimum, they should provide detailed assessment of the performance of this tool against validated methods. At this point, all we can see from the data is that  one method gives one reading and the other gives a very different one. The correlation coefficient of 0.497, although statistically significant is actually very poor. Besides, this is not the appropriate way of evaluating the diagnostic accuracy of two different methods, the authors should have used Bland-Altman plots with a lot stricter than the customary 30% SD intervals.

5. The authors used multivariate regression to evaluate if CLs and SLs are independent predictors of mortality. There is no mention, how the model was built and what was the model fit to the population. This reviewer can only assume that the authors used the result of the univariate analysis to add variables to their model. This method, although commonly performed is very highly problematic. It is prone to producing spurious associations, and missing genuine ones (GW Sun, TL Shook and GL Kay.  Inappropriate use of bivariable analysis to screen risk factors for use in multivariable analysis.  Journal of Clinical Epidemiology 49(8):907-916, 1996).   Similarly, as Harrell explains (FE Harrell Jr. (2001). Regression Modeling Strategies. New York, Springer: 56-60), stepwise variable selection violates numerous fundamental assumptions of statistical analysis that can easily give spurious results.  Evidence of the dangers of seeking parsimony with these methods can be seen in these two simulation studies:  PC Austin and JV Tu, J Clin Epidemiol 57(11):1138-1146, 2004; S Derksen and HJ Keselman.  Br J Math and Stat Psychology 45:265-282, 1992.  If parsimony is necessary because of concerns about overfitting (which is an issue with small data sets, and is a particular issue for the small # of deaths in this cohort), or other concerns, it should be achieved by choosing the most likely covariates based on pre-existing knowledge or first principles, or by the kind of advanced variable reduction strategies discussed in some detail in Harrell's book.

6. Given these significant methodological issues, I don't feel the data presented supports the assertions in the Discussion. In the first paragraph, the authors talk about septic shock, which is NOT the pathophysiological process seen in their patients, given that the rate of vasopressor support was very low overall. I understand that they try to wrap their conceptual framework around the results, however as stated above, I think they should look at their data first, then try to see if it support their hypothesis. The whole first paragraph of the Discussion is irrelevant to this study. The only relevant sentence is the following: ...CLs assay (and the related cut-off) has not been validated yet...

7. The authors do not provide explanation to what the +LR or -LR abbreviation means in the results, yet try to assert that these are somewhat presenting a "better" predictive value of CLs over SLs. Furthermore the higher OR doesn't mean one  test is "better" at diagnosis, it only means there is a potentially stronger relationship with outcome. However, given the general inadequacy of the multivariate model, I have significant doubts about the veracity of this. 

8. In the Conclusion, the authors say that CLs rise earlier than SLs. They have not provided any data to show this so this strong assertion is potentially misleading. Please delete.

Author Response

Please find attached the file with a thorough point-by-point reply to each comment raised by the Reviewer.

Reviewer 2 Report

Thank you for giving me the opportunity to review this article. This study dealt with the performance of capillary lactate compared to serum lactate in patients with suspected sepsis/septic shock admitted in one emergency department. This study is interesting, however could be improved. 

Here, my main remarks. 

first, as a general comment, in medical articles, we are used to have the structure introduction/materials and methods/results/ discussion/ conclusion. Would it be possible to have this structure? 

Then the introduction seemed relatively complete and well introduce the concept of sepsis. 

material and methods. 

it would have been interesting, instead of presenting the results with the Fagan’s nomograms to have ROC curves and associated AUC for the prediction of death a day 2 and 7 with the comparison of the curves for CL, SL, SOFA and NEWs

it would be interesting to have the levels of CL and SL in included patients depending on their 'no sepsis', 'sepsis' and septic shock status', . 

Would it be possible to report results for day-28 mortality and also prediction for ICU admission. 

 I suppose you get only one measurement of CL and SL for each patients, and are not able to report the predictive value of the kinetics of CL and SL on your outcomes? it could be had into the limits of the manuscrit. 

Did you have missing values? 

Sub group analyses depending on the source of sepsis would be interesting. 

Discussion: does CL has been already compared to other tools assessing microcirculation? 

Table S2: could you precise, what are the OR corresponding for. 

Author Response

(The authors gave the same response as above.)

Reviewer 3 Report

Abstract: can you make a statement regarding the advantage of a CL measurement over a SL measurement?

Line 15: Write out capillary lactates before (CL) since that is done for serum lactates in the same sentence.

Lines 20 and 21: Not clear what is meant in this sentence and the other sentence like it in the abstract: “Patients deceased within 48 hours had higher CLs (19.3 vs. 5 mmol/L, p<0.001) than SLs (6.5 vs. 1.1 mmol/L, p=0.001).”  Was expecting two numbers to be compared (CLs vs. SLs), but instead there are a total of 4.  What is meant by the 19.3 vs. 5 mmol/L values? 

Line 32: Why is the mortality rate expected to increase?

Line 48: Instead of evokes, use leads to?

Table 1: Please provide additional annotation throughout Table 1.  For example what does “Male 79 (42.5)” mean?  Does this mean that 79 males survived and the percentage of those that survived was 42.5 (I get 79/184 = 42.9)?  It would be helpful to provide a better description of what most of the clinical/lab features are and what the numbers represent.

Table 1: Is there a corresponding row for females?

Table 1: What is meant by altered mental status?  What is the number and what is the number in parentheses?

Line 204: consider changing, “First, the prospective analysis…” to “First, this is a prospective analysis…”

In lines 209-211, can you further explain how a CL approach would limit physician biases?

How does the LactatePro instrument work (what is the principle behind how this operates – enzymatic reaction, etc.?) and what range of lactate values can be accurately measured?  Is calibration of this instrument needed?  Please provide a brief description in the manuscript.

When were the SL samples taken – upon patient arrival at the same time that the CL samples were taken (maybe specify this in lines 238-239)? 

Can the authors address how much time a CL vs SL measurement saves?  It seems like CL measurements have quantitative advantages; can you explain what those are?

Author Response

(The authors gave the same response as above.)

Round 2

Reviewer 1 Report

Thank you for providing some more information on the data in the manuscript. Some of my questions have been addressed, however there are several areas remain where the authors and this reviewer is at odds.

1. The introduction has improved and easier to follow.

2. Thank you for providing the information on the accuracy of CL and SL values. Looking at the Bland-Altman plot, I can't fail to notice the significant outliers when lactate levels are high. Indeed, from the plot it is clear that at low lactate levels the two measurements are closely aligned, however as lactate levels rise, the variability increases significantly. This could have important clinical considerations and although the authors have partially addressed this limitation in the new version of the manuscript, I'd urge them to spell it out clearly.

3. The authors didn't managed to convince me, that their multivariate model is built appropriately. Were the variables entered into the model based on prior knowledge? In which case why they have left out the consistently show independent predictors such as age, comorbidity and frailty? How can they assert that there was no multicollinearity between the factors, when they also assert that CLs and SLs are highly correlated? I don't understand how they used linear regression analysis, when the outcome they were trying to predict was a binary categorical variable. I'd seriously consider getting advice from an expert in medical statistics as the current modelling strategy does not make sense. I appreciate, that the authors want to reduce the number of tested variables to minimum, but how they come up with a list can cause significant repercussions in their results.

It would also be imperative to show the goodness-of-fit tests for the final model they employ, so the readers can appreciate if it would fit their own patient population.

4. I still feel that the first paragraph in the Discussion is completely superfluous. It is customary, that at the beginning of the Discussion the authors to state the main findings in max 2 sentences, then they put them into context. This latter part doesn't arrive until the second paragraph of the current Discussion. 

5. The authors put forward a hypothesis in the Discussion that the different correlations and lack of correlations between blood pressure and lactate values might mean CL rises before SL does. Unfortunately they selectively reporting their own results, as even the currently reported multivariate model shows that both SLs and CLs are independent predictor of outcome at 48 hours. They write: "...although only CLs is a strong predictor of short-term mortality independently from hypotension development." however they have not provided any data to back this statement as they have not included hypotension in their model. I continue to struggle to understand what is the message the authors would like to convey in this paragraph?

Reviewer 2 Report

The authors took into account most of the suggestion in this modified version of the manuscrit. I have no more remark.